# Flatness-Aware Prompt Selection
# Improves Accuracy and Sample Efficiency

**Lingfeng Shen**[♡]     **Weiting Tan**[♡]     **Boyuan Zheng**     **Daniel Khashabi**
Center for Language and Speech Processing  and  Computer Science Department
Johns Hopkins University, Baltimore MD
{lshen30, wtan12, bzheng12, danielk}@jhu.edu

## Abstract

With the growing capabilities of large language models, prompting them has become the dominant way to access them. This has motivated the development of strategies for automatically selecting effective language prompts. In this paper, we introduce PFLAT (prompt flatness), a new metric to quantify the expected utility of a language prompt. This metric is inspired by *flatness* regularization in statistical learning that quantifies the robustness of the model towards its parameter perturbations. We provide theoretical foundations for this metric and its relationship with other prompt selection metrics, providing a comprehensive understanding of existing methods. Empirically, we show that combining PFLAT with existing metrics improves  both performance and sample efficiency. Our metric outperforms the previous prompt selection metrics with an average increase of 10% in Pearson correlation across 6 classification benchmarks, and the prompt selected by our metric gains 5% higher accuracy than previous metrics across the benchmarks.[1]

## 1   Introduction

Manually "engineering" prompts for large language models (LLMs) have been shown to lead to tremendous performance gains and have been a subject of intense study in recent years (Schick and Schütze, 2021a; Reynolds and McDonell, 2021; Mishra et al., 2022). However, the task of prompt engineering can be challenging due to the difficulty in determining the effectiveness of a prompt solely based on its raw text form. Consequently, this process is typically carried out manually, which can be laborious and time-intensive. In particular, LLMs may produce vastly different predictive distributions for two seemingly comparable prompts, despite their semantic similarity (Mishra et al., 2022).

---

♡ Equal contribution.

[1]The code is accessible here: https://github.com/shadowkiller33/flatness.

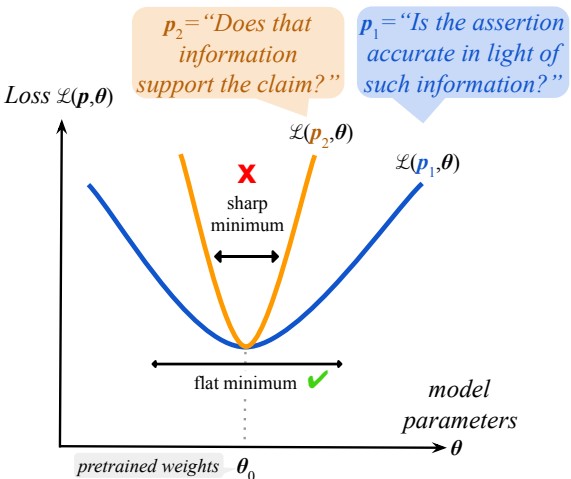

Figure 1: We show that *prompt flatness* is an effective indicator of a prompt's performance on an LLM. For example, if two prompts $p_1$, $p_2$ incurs the same loss on an LLM parameterized by $\theta_0$, i.e., $\mathcal{L}(p_1, \theta_0) = \mathcal{L}(p_2, \theta_0)$, we find that the one with a flatter loss landscape of LLM parameters ($p_1$, in this visualization) is better.

This phenomenon results in an unexpectedly high level of variability. (Jiang et al., 2020; Perez et al., 2021; Elazar et al., 2021).

In response to such difficulties, recent works propose metrics for automatic prompt selection. Notably, Sorensen et al. (2022) introduces *Mutual Information (MI)* to quantify the shared information between prediction and inputs. Further, Chen et al. (2022) introduces *Sensitivity (SEN)* to quantify model receptiveness to textual perturbations of the input prompts. Despite such metrics' empirical effectiveness, the underlying principles that enable them are not well understood.

This motivates the following questions: ($RQ_1$) What makes the existing methods for prompt selection effective? ($RQ_2$) How are these existing methods connected? ($RQ_3$) Are there any new metrics complementary to the existing ones?

To address the questions above, we study existing methods from an optimization perspective. The

objective $\mathcal{L}(p, \mathcal{D}, \theta_0)$ quantifies the performance of an LLM (parameterized by $\theta_0$) on labeled data $\mathcal{D}$ and a prompt $p$ appended to the dataset inputs. Prompt selection is in effect an optimization on $\mathcal{L}(p, \mathcal{D}, \theta_0)$ as a function of different choices of $p$. The challenge is that, in practice, there are few labeled data $\mathcal{D}$ (Perez et al., 2021), which would make $\mathcal{L}(.)$ an unreliable measure for selecting effective prompts. We show that the existing prompt selection metrics (MI and SEN) (Sorensen et al., 2022; Chen et al., 2022) approximate the objective function $\mathcal{L}$, and therefore, act as its *surrogates*. This addresses $(RQ_1)$ and $(RQ_2)$ above.

Additionally, to address $(RQ_3)$ we borrow ideas from statistical learning on flatness-aware optimization (Hochreiter and Schmidhuber, 1994; Keskar et al., 2017). We introduce *Prompt Flatness* (PFLAT), a metric that quantifies $\mathcal{L}$'s sensitivity to small perturbations in LLMs parameters, when conditioned on a prompt (see Figure 1 for intuitions). Our results indicate that prompts with higher flatness generally lead to better accuracy.

Our formal derivations also show that PFLAT is distinct from and complementary to prior metrics such as MI and SEN. Our empirical results (§3) on six classification benchmarks and four different model sizes also confirm our theoretical intuition. For example, combining PFLAT and MI improves the downstream performance by 6% accuracy over the prompts selected by MI only. Similarly, combining PFLAT and SEN improves the downstream performance by 9% accuracy over prompt selected by SEN only. Additionally, using PFLAT substantially improves sample efficiency, an important feature of low-resource scenarios.

In summary, our contributions are: (a) We propose a formal optimization framework that unifies several existing prompt selection metrics such as MI and SEN. (b) Enabled by our formalism, we introduce PFLAT, a metric for selecting prompts that is more robust to LLMs' parametric perturbations. (c) We conduct comprehensive experiments and the results demonstrate the effectiveness of our method for prompt selection.

## 2 Prompt Selection via Flatness

We start by introducing the necessary background and the notational convention (§2.1), then introduce our proposed metric, PFLAT (§2.2), followed by a discussion of its relation to other existing prompt selection metrics (§2.3).

### 2.1 Background and Setup

**Notation.** We cast prompt selection into an optimization problem. We are provided with a pretrained language model $f$ with parameters $\theta \in \mathbb{R}^m$ which maps each input natural language instance $\boldsymbol{x}$ to $f_\theta(\boldsymbol{x}) \in [0, 1]^{|V|}$, a distribution over the label set $V$. We are also given input-output pairs $\mathcal{D} = \{(\boldsymbol{x}, \boldsymbol{y})\}$, where $\boldsymbol{y}$ is a one-hot label.

**Prompt selection.** Given a language model $f$, we seek to minimize the following empirical risk, also called prompt loss in this paper:

$$\mathcal{L}(p, \mathcal{D}, \theta) = \frac{1}{|\mathcal{D}|} \sum_{(x,y) \in \mathcal{D}} \ell(f_\theta(p \circ \boldsymbol{x}), \boldsymbol{y}),$$

where $p \circ x$ is the string combination of a prompt $p$ to input $x$, and $\ell$ is an appropriate loss such as cross-entropy that quantifies the gap between gold label $\boldsymbol{y}$ and predicted distribution $f_\theta(p \circ \boldsymbol{x})$.

In the classic machine learning literature, it is customary to minimize empirical risk $\mathcal{L}(p, \mathcal{D}, \theta)$ with respect to the parameters of the underlying model $\theta$. However, the recent developments in LLMs (Radford et al., 2019; Brown et al., 2020) have resulted in an alternative that involves optimization concerning the choice of prompt $p$:

$$\hat{p} = \underset{p \in \mathcal{P}}{\arg\min}\, \mathcal{L}(p, \mathcal{D}, \theta), \tag{1}$$

given a collection of natural language prompts $\mathcal{P}$ that are "engineered" by domain experts (Schick and Schütze, 2021b,a; Mishra et al., 2022)

### 2.2 Prompt Selection via Flatness

Our work draws inspiration from classic machine learning, where studies have demonstrated that using loss *flatness* in model selection leads to improved performance and generalization (Foret et al., 2020; Baldassi et al., 2020; Zheng et al., 2021; Stutz et al., 2021; Andriushchenko and Flammarion, 2022). In this prior literature, the optimization is performed with respect to model parameters $\theta$. Conversely, in the modern NLP literature, the parameters of LLMs are set once they are pre-trained, and further optimization is achieved through input prompts As a result, it remains to be seen whether the findings from classic machine learning literature will translate to prompting LLMs.

**Robust prompt selection objective.** We start with the formal definition of flatness. Specifically,

the goal is to select parameters that are robust to parameter perturbations:

$$\bar{\mathcal{L}}(p, \mathcal{D}, \theta) = \max_{\|\epsilon\| < r} \mathcal{L}(p, \mathcal{D}, \theta + \epsilon), \quad (2)$$

$$\hat{p} = \arg\min_{p \in \mathcal{P}} \bar{\mathcal{L}}(p, \mathcal{D}, \theta), \quad (3)$$

where $\epsilon$ is a small perturbation added to model parameters $\theta$. The inner optimization quantifies the worst-case loss upon a small perturbation of the model parameter from its default value, where the perturbations are contained within a small algebraic ball, $\|\epsilon\| < r$. The overall objective is a minimax optimization (Zheng et al., 2021; Stutz et al., 2021; Baldassi et al., 2020) i.e., selecting the best prompt $p$ with the smallest worst loss under small perturbations. Note that this is a strict generalization of the standard prompt selection objective in Equation 1.

**Flatness definition.** Since Equation 2 is a nontrivial saddle-point optimization problem, previous work (Zhao et al., 2022; Zhang et al., 2023b) has approximated it with the gradient norm of loss function:

$$\bar{\mathcal{L}}(p, \mathcal{D}, \theta) \approx \mathcal{L}(p, \mathcal{D}, \theta) + \alpha \mathcal{F}(p, \mathcal{D}, \theta) \quad (4)$$

$$\mathcal{F}(p, \mathcal{D}, \theta) \triangleq \|\nabla_\theta \mathcal{L}(p, \mathcal{D}, \theta)\|_2, \quad (5)$$

where $\mathcal{F}(p, \mathcal{D}, \theta)$ is the accurate analytical definition of *flatness* the loss function $\mathcal{L}(.)$. Intuitively, it quantifies how resilient it is against small perturbations in parameter space $\theta$.

The calculation of $\mathcal{F}$ requires (1) gradient computation of the loss $\mathcal{L}$ and (2) ground-truth labels which may not be available. To circumvent these challenges, we introduce an approximation of $\mathcal{F}$.

**An efficient surrogate for flatness.** Here provide an approximate definition of flatness ($\mathcal{F}$ in Equation 5) that does not depend on instance labels. Our new metric, PFLAT quantifies the amount of changes in LLM confidence values upon perturbations in its parameters:

$$\text{PFLAT}(p, \mathcal{D}_X, \theta) =$$
$$\frac{1}{|\mathcal{D}_X|} \sum_{\boldsymbol{x} \in \mathcal{D}_X} \mathbf{E}_{\epsilon_1, \epsilon_2} \Big[ g(\epsilon_1) - g(\epsilon_2) \Big], \quad (6)$$

where $g(\epsilon) \triangleq \ell\Big( f_\theta(p \circ \boldsymbol{x}), f_{\theta+\epsilon}(p \circ \boldsymbol{x}) \Big)$ and $\epsilon_1, \epsilon_2$ are sampled from a Gaussian distribution $\mathcal{N}(0, \sigma^2)$ with its variance $\sigma^2$ determining the perturbation magnitude. Furthermore, $\mathcal{D}_X = \{\boldsymbol{x}\}$ refers to the

input instances only (no labels). Intuitively, higher PFLAT means higher sensitivity towards perturbation in model parameter, indicating that the given input prompt, instances, and the model parameters have formed a sharper minimum. The formal connection between PFLAT and $\mathcal{F}$ is deferred to Appendix B.

Although the precise computation of PFLAT demands numerous Gaussian samples, practically, approximating it with few samples suffice for a reasonable PFLATestimate. We'll demonstrate this in the experiments (§4).

**Putting it together.** Incorporating our PFLAT metric (Equation 6) in robust prompt selection objective (Equation 4) we get the following:

$$\bar{\mathcal{L}}(p, \mathcal{D}, \theta) \approx \mathcal{L}(p, \mathcal{D}, \theta) + \alpha \cdot \text{PFLAT}(p, \mathcal{D}, \theta), \quad (7)$$

where $\alpha$ is a scalar hyperparameter. In our experiments, we select the prompt with the smallest $\bar{\mathcal{L}}$ and show that such prompts have better quality than those selected only by MI or SEN. For emphasis, this equation shows that for robust prompt selection according to $\bar{\mathcal{L}}$, it is not enough to use PFLAT alone. It should be used in conjunction to $\mathcal{L}$ or its approximations (discussed in the next section). We show this point empirically in Section 3. The only reason that our metric is not fully zero-shot is that the hyper-parameter $\alpha$ has to be selected according to a few examples of a held-out set.

## 2.3 Relation to Prior Prompt Metrics

We show that prompt selection through existing methods such as MI (Sorensen et al., 2022) and SEN (Chen et al., 2022) is approximately equivalent to minimizing prompt loss $\mathcal{L}(p, \mathcal{D}, \theta)$, as shown in Equation 5. Therefore, they can be viewed as surrogates to $\mathcal{L}(.)$. Formally, we provide the gap between prompt loss and its surrogates (e.g., MI and SEN) which is determined by the difference (e.g., KL divergence) between a model's predictions and the ground-truth labels.

**Mutual Information.** Sorensen et al. (2022) propose to pick prompts that maximize the mutual information between model input and the prediction.

**Proposition 1.** *Mutual information* $\text{MI}(p, \mathcal{D}, \theta)$ *is a surrogate loss for prompt loss* $\mathcal{L}(p, \mathcal{D}, \theta)$ *with a*

*gap quantitatively defined as follows:*

$$\text{MI}(p, \mathcal{D}, \theta) - \mathcal{L}(p, \mathcal{D}, \theta)$$
$$= c + \frac{1}{|\mathcal{D}|} \sum_{(\boldsymbol{x}, \boldsymbol{y}) \in \mathcal{D}} \text{KL}(f_\theta(\boldsymbol{x} \circ p) || \boldsymbol{y}),$$

*where $c$ is a constant $c = H(f_\theta(\boldsymbol{x} \circ p))$ that does not depend on prompt $p$.* KL *refers to KL divergence.*

**Sensitivity.** Give a prompt $p$, Chen et al. (2022) utilizes the sensitivity of model prediction towards the textual perturbation in $p$.

**Proposition 2.** *Sensitivity* $\text{SEN}(p, \mathcal{D}, \theta)$ *is a surrogate loss for prompt loss* $\mathcal{L}(p, \mathcal{D}, \theta)$ *with a gap defined as follows:*

$$\text{SEN}(p, \mathcal{D}, \theta) - \mathcal{L}(p, \mathcal{D}, \theta)$$
$$= \frac{1}{|\mathcal{D}|} \sum_{(x, y) \in \mathcal{D}} \mathbf{E}_{p'} \left[ \ell_{01}(f_\theta(x \circ p'), y) \right],$$

*where $p'$ and $\ell_{01}$ refer to the perturbed prompt and 0-1 loss, and $\mathbf{E}_{p'}$ is an expectation (average) over different choices of perturbed prompts $p'$*

The detailed analyses are deferred to Appendix A. These derivations show that selecting prompts based on MI and Sen is approximately selecting the prompts with the smallest prompt loss, which shows their connections and explains why they are effective for prompt selection.

**Complementarity to PFLAT.** A corollary of Proposition 1,2 is that prompt-selection metrics such as MI (Sorensen et al., 2022) and SEN (Chen et al., 2022) are surrogates for prompt loss, which are complementary to PFLAT, for the purpose of robust prompt selection (Equation 2). To see this, it is enough to go back to Equation 7, which shows how robust prompt selection decomposes into PFLAT and $\mathcal{L}$. Finally, as we see, $\mathcal{L}$ is approximated by SEN and MI, which concludes the argument.

## 3 Experiments

We conduct extensive experiments to assess the effectiveness of prompt selection metrics.

**Experimental setup.** We experiment with a variety of classification benchmarks: AGNews (Zhang et al., 2015), CB (De Marneffe et al., 2019), DBpedia (Zhang et al., 2015), SST-2 (Socher et al., 2013), RTE (Dagan et al., 2005), and TREC (Voorhees

and Tice, 2000). We choose four different GPT-2: `base`, `medium`, `large`, and `xl` sizes.[2]

**Held-out set for $\alpha$ hyperparameter.** For each dataset, we create a small dev-set by randomly selecting 8 labeled sentences per class to tune the value of $\alpha$.

**Implementation.** We prepare 20 human-written instructions by the authors (included in Appendix F) appended by random demonstrations for each task. The number of demonstrations is set as 5, which matches the settings in Sorensen et al. (2022) for a fair comparison. We use MI, SEN, PFLAT, and their combinations for comparison. The results are averaged on three random seeds. We estimate PFLAT (Equation 6) via 5 random Gaussian perturbations of LLM parameters with variance $\sigma^2$ set to `1e-4`. Later, we offer an assessment of the influence of this estimation (§4.4).

**Evaluation metrics.** We use two metric families: *Correlation with accuracy:* The first category measures the alignment between prompt selection metrics (including our proposed metric) and the downstream accuracy of each prompt. This evaluation contrasts the relative quality of prompts based on their accuracy with their prompt-selection accuracy. Specifically, for each prompt, we compute the prompt selection metric score (MI, MI + PFLAT, which uses only task inputs) and the prompt's accuracy on the test set. Given a collection of such paired numbers, we compute their correlation. A high correlation indicates that this prompt-selection metric can serve as a "surrogate" (proxy) for selecting the most accurate prompt, bypassing the direct maximization of accuracy which often demands extra held-out labeled data.
*Ranking evaluation:* Since correlations are sensitive and brittle to outliers (Anscombe, 1973), we further use different metrics for best-performance prompt retrieval. Specifically, we use NDCG@1 (Järvelin, 2000), NDCG@3, and Rate. NDCG is a common metric for ranking quality in information retrieval. Here, we take prompts' performance as their quality score for NDCG. We denote the prompt selected by metric (e.g., highest MI or lowest SEN) as $\hat{p}$, and `Rate` is defined as follows:

$$\text{Rate} = \frac{\text{Performance}(\hat{p})}{\text{Performance}(p_o)}, \quad (8)$$

---

[2]The models are accessible at `https://huggingface.co/gpt2`.

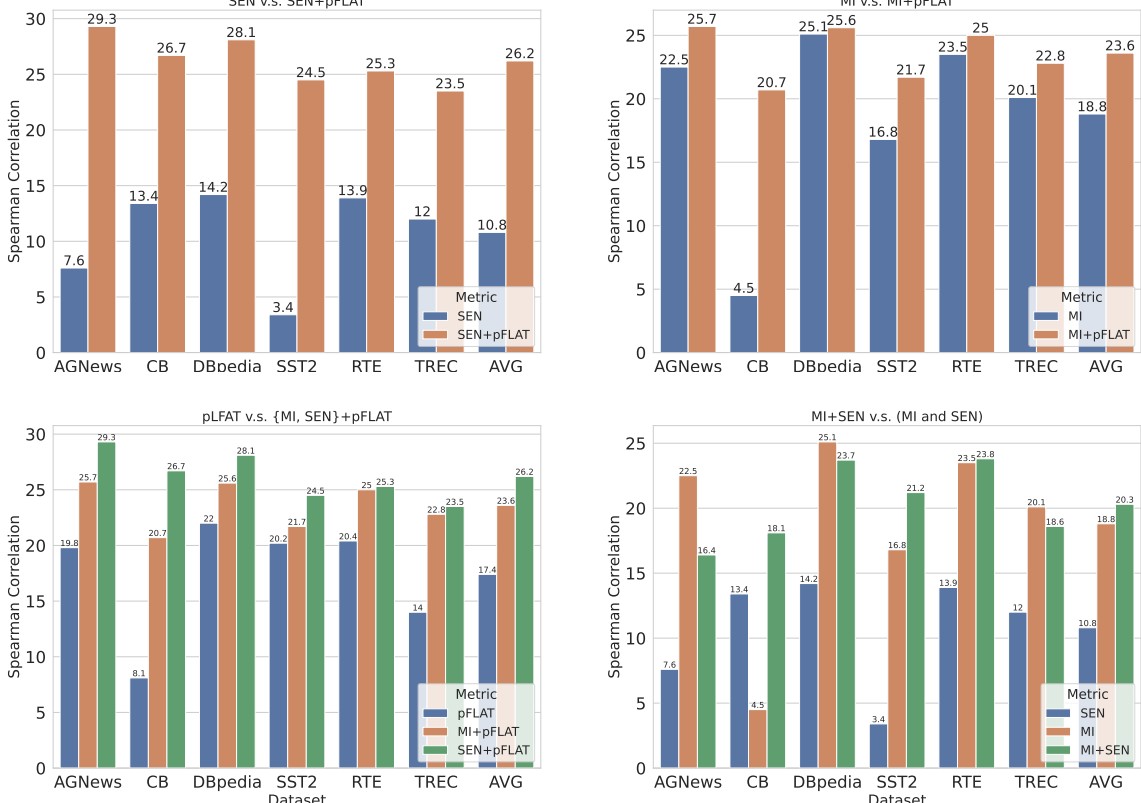

Figure 2: Results of correlation evaluation across six datasets and their average (AVG). First row: SEN vs SEN+PFLAT and MI vs MI+PFLAT show that **flatness brings consistent improvements over existing metrics**. Bottom left: From PFLAT vs MI+PFLAT, flatness does not perform well when applied alone, as expected. Bottom right: MI+SEN vs MIcomparison shows that combining SEN and MI brings limited improvement.

where $p_o$ refers to the prompt that achieves the best performance on the task. Intuitively, `Rate` reflects the performance of the selected prompt compared to the best prompt, and it is a real-valued number between 0 and 1. A larger `Rate` corresponds to a better selected prompt $\hat{p}$.

**Flatness is complementary to MI and SEN.** The correlation results are in Figure 2 (detailed numbers in Appendix C). Figure 2 (first row) shows that correlations are higher for MI+PFLAT and SEN+PFLAT than for metrics without PFLAT. In other words, combining existing (MI or SEN) with flatness results in a more effective prompt selection metric that correlates better with test accuracy.

We find similar results in the ranking evaluation illustrated in Figure 3 (full results in Appendix D). In all benchmarks, metrics incorporating flatness generally surpass those without it, highlighting the importance of utilizing prompt flatness in the prompt selection process.

**Flatness alone does not help.** As shown in Figure 2, SEN+PFLAT, MI+PFLAT or MI generally outperforms PFLAT, these results show the importance of combining prompt loss. Without prompt

loss, prompt flatness on itself is insufficient to reflect prompt quality. Such results also stress the importance of combining prompt loss and flatness.

## 4  Further Analysis

### 4.1  Continuous Prompt Selection

In addition to text form (discrete) prompt, we also test out the effectiveness of flatness for continuous prompt optimization (also known as 'prefix-tuning'). Like the earlier result, introducing flatness to prefix-tuning also improves model performance.

| Method | SST-2 | AGNews | SNLI |
|---|---|---|---|
| w/o Flatness | 92.5 (0.1) | 86.4 (0.2) | 72.5 (0.2) |
| w/ Flatness | **93.1** (0.1) | **87.3** (0.1) | **73.3** (0.2) |

Table 1: Performance for prefix tuning with flatness and w/o flatness in a mean (standard-deviation) form. It is observed that **leveraging flatness in continuous prompt tuning brings improvements to performance**. Stronger numbers for each dataset are marked **bold**.

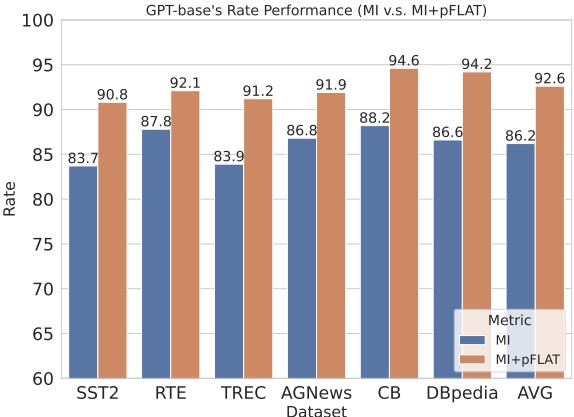
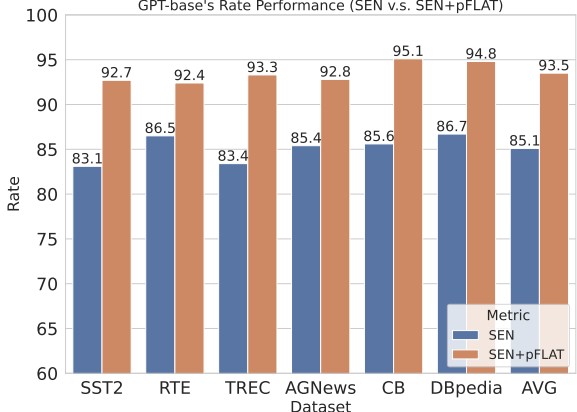

Figure 3: Rate(reflecting the ability to select better prompts) computed for prompt selection across six datasets and their average performance, using GPT2-base model. We can see that combining **flatness with existing metrics (MI+PFLAT or SEN+PFLAT) is consistently better than not using PFLAT**.

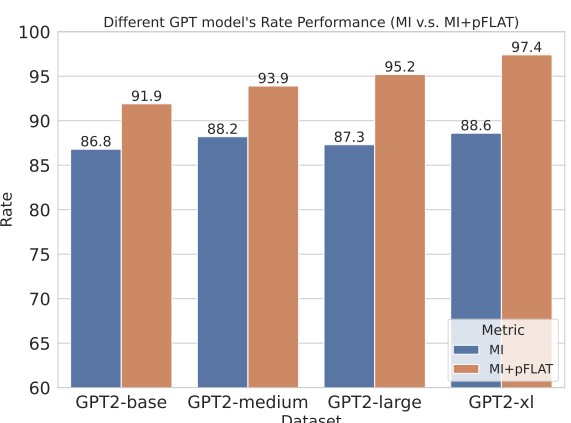
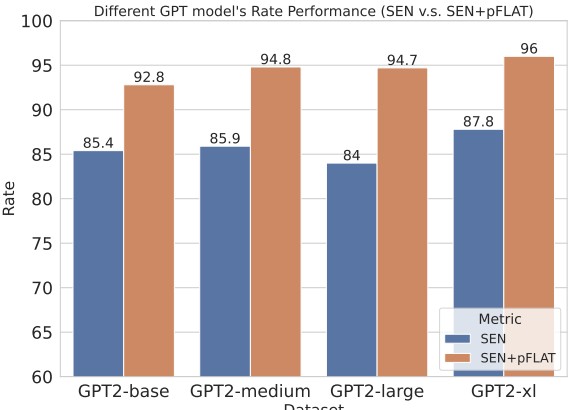

Figure 4: Rate (reflecting the ability to select better prompts) evaluation computed prompt selection across four model sizes, evaluated on the AGNews dataset. Combining prompt loss and flatness (MI+PFLAT or SEN+PFLAT) is consistently better than MI/Sen alone across different model types. More detailed results are deferred to Appendix D.

**Experimental setup.** We following prefix-tuning setup of Li and Liang (2021) and consider three text classification benchmarks in our experiments: SST-2 (Socher et al., 2013), AGNews (Zhang et al., 2015), and SNLI (Bowman et al., 2015). We use the GPT2-medium as the model and set prefix length to 10 tokens for all prefix-tuning experiments. We train 30 epochs for SST-2 and 25 epochs for AGNews and SNLI, as suggested in Yang and Liu (2021).

**Implementation of flatness-aware prefix-tuning.** To introduce flatness into prefix tuning, we leverage sharpness-aware optimizer SAM (Foret et al., 2020) for model optimization. We use Adam (Kingma and Ba, 2015) as our optimizer in the counterpart without flatness. Specifically, both cases use the same learning rate 5e-5.

**Results.** As shown in Table 1, prefix-tuning with flatness achieves better performance than without flatness. Such results show that **flatter continuous prompts bring better performance**, which matches our conclusions on discrete prompts.

### 4.2 Influence of Model Size

We investigate the effects of model size in our methods. As shown in Figure 4, as the model size increases the gap between the two metrics (e.g., MI vs MI+PFLAT) measured in terms of Rate generally increases, indicating an increasing gain from adding PFLAT to existing prompt selection for larger models.

### 4.3 Impact on Sample Efficiency

If there is enough labeled data, a reasonable approach for prompt selection is based on the ac-

curacy of the prompt on a labeled development set (we name this baseline method "acc"). Thus, a natural question concerning practicality arises: how does our method compare to prompt selection based on the accuracy of limited labeled examples? To perform the comparison, we select $N$ labeled data from the AGNews dataset and evaluate `Rate` (Equation 8) for both "acc" baseline and our method (MI/SEN + PFLAT).

Based on the results in Figure 5, we observe that with little data available, our methods select a far better prompt than the "acc" baseline, allowing performance gains in low-data scenarios. This can be attributed to the fact that when the dataset is small, there may be a significant distribution shift between the development and test sets. However, our methods, MI/Sen/PFLAT, provide signals beyond labeled data and thus more resilient to such distribution shifts. Unsurprisingly, when data size grows, the gap between our method and the "dev" baseline decreases since the distribution shift issue is mitigated by increasing the size of the dev set. In conclusion, our metrics are more advantageous than development set accuracy for prompt selection in low-resource scenarios.

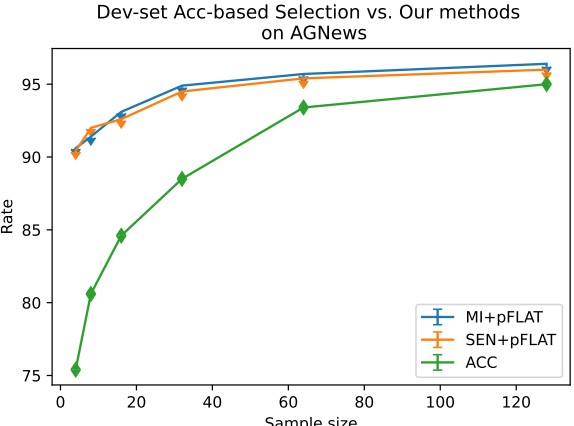

Figure 5: For development sets with varying sizes $n = 16, 32, \cdots, 512$, the devset-acc based method (green line) selects prompts based on the accuracy of prompts on $n$ devset samples. On the other hand, our metric (MI+PFLAT and SEN+PFLAT) also use $n$ samples and achieve better performance under low-resource scenarios ($n < 500$).

### 4.4 Estimation of PFLAT

In our implementation of PFLAT (Equation 7), there are two factors that affect PFLAT: the sampling number $N$ and the perturbation size $\sigma^2$. We explore their effects in this part.

As noted earlier, we compute prompt flatness by sampling $\epsilon$ from a standard Gaussian distribution $\mathcal{N}(0, \sigma^2)$. Since the computational cost of this estimate is proportional to the sample size $N$, the choice of $N$ is crucial for our efficiency. Figure 7 shows the results of an experiment showing the trade-off between $N$ and estimation quality. The results indicate that $N \approx 5$ is sufficient to provide reliable estimates for PFLAT.

Likewise, we investigate the impact of $\sigma^2$ on the estimate. The results in Figure 6 (a, b) indicate that the optimal perturbation size is around `1e-4`. When the perturbation size increases after `1e-4`, the estimation error also increases.

## 5 Related Work

**Prompt selection and engineering.** Performance of LLMs is highly sensitive to their prompt prefix, including the ordering of demonstrations (Lu et al., 2022) or framing of the instructions (Mishra et al., 2022). This has motivated work prompt selection, such as the ones discussed in this work (Chen et al., 2022; Sorensen et al., 2022). Beyond quantifying prompts' effectiveness, the literature has explored alternative ways to address LLMs' brittleness, such as chain-of-thoughts prompting (Kojima et al., 2022), LLM self-consistency (Wang et al., 2022a) and complexity (Fu et al., 2022). Our optimization-based framework does not cover these classes of prompt engineering, which we hope future work will address.

**Algorithmic prompt generation.** Several prior works focus on generating effective prompts to solve a given task via an LLM. Examples are RLPrompt (Deng et al., 2022), GrIPs (Prasad et al., 2023), and Tempera (Zhang et al., 2023a). While these works primarily focused on *generating* prompts with high performance for *prompt-tuning*, our goal is to *identify* effective prompts for a pool of candidate prompts that is beneficial for *in-context learning*. In particular, within the context of Appendix E, a comparative analysis is conducted to examine the in-context learning performance of prompts generated by these approaches. The results reveal that prompts deemed suitable for fine-tuning exhibit sub-optimal performance in terms of in-context learning.

Besides, the ability to *generate* prompts inevitably involves model tuning via setups like Reinforcement Learning which incurs an additional computational cost. More importantly, the qual-

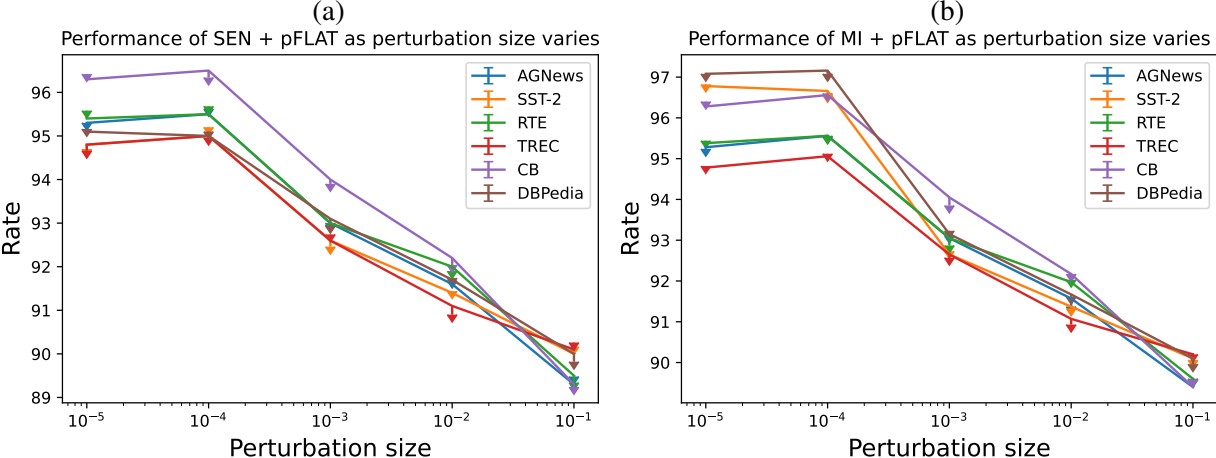

Figure 6: (a) `Rate` of SEN+PFLAT as perturbation size varies. The optimal $\epsilon$ is around `1e-5`, as $\epsilon$ enlarges, the performance of SEN+PFLAT continues to degrade. (b) `Rate` of MI+PFLAT as perturbation size varies. The trend in MI+PFLAT is similar to (b).

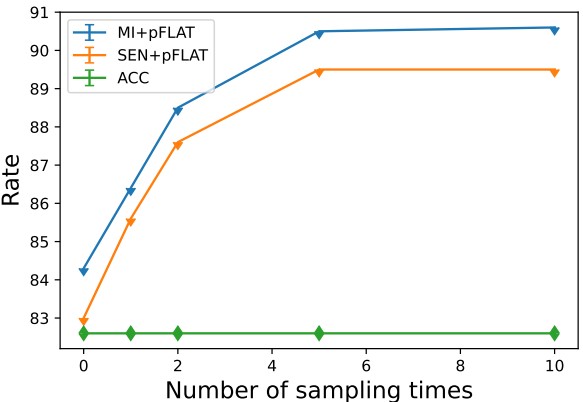

Figure 7: The trade-off between performance and sampling number $N$ in PFLAT's computation procedure. We can observe that the proper $N$ is around 5.

ity of the generated prompts depends on the task's domain. When confronted with Out-of-Domain (OOD) tasks, these approaches tend to generate nonsensical prompts.

**Continuous prompts.** Beyond language (discrete) prompts, we show that our results also apply to continuous prompts. In contrast to manually creating discrete prompts, one can optimize continuous prompts in embedding space, yielding better results (Lester et al., 2021; Li and Liang, 2021; Zhang et al., 2022; Gu et al., 2022; Lang et al., 2022; He et al., 2022). Despite higher accuracy, continuous prompt optimization is only applicable to LLMs that are publicly accessible. Besides, there is no evidence that continuous prompts are interpretable (Khashabi et al., 2022), making it

challenging to transfer insights from prompts that work well for one task to another.

**Flatness-aware language modeling** Previous works (Liu et al., 2023; Mehta et al., 2021) showed that flatness-aware optimization can enhance the generalization of LLM during pre-training, even if the training loss is the same. Na et al (Na et al., 2022) demonstrated that flatness-aware training increases the compression rate. Wang et al(Wang et al., 2022b) showed the advantages of flatness in training encoder-only models.

**Model calibration and robustness analysis.** Model calibration focuses on adjusting LLMs' predictions to reflect human uncertainty (Holtzman et al., 2021; Zhao et al., 2021; Jiang et al., 2022). Calibration is related to our work as a well-calibrated LLM's confidence could be used for prompt selection. However, calibration algorithms have remained domain/task-specific so far, restricting their applicability to the problem discussed in this paper.

## 6 Conclusion

We developed a theoretical framework for prompt selection techniques that merges prompt loss and flatness, enabling the integration of previous studies to elucidate their distinctions and efficacy. Through extensive experimentation, we demonstrated the effectiveness of our proposed flatness-based metric when used in conjunction with existing ones. Our research offers valuable insights and directions for future investigations in effective

prompt engineering.

## Limitation

The limitations of this study can be outlined as follows: (1) Our paper assesses the methods based on classification tasks, but they can potentially be applied to generation tasks in the future. (2) Our framework presumes that the provided collection of candidate prompts is all coherent and fluent for the intended task, despite the possibility of yielding varying results. (3) Our approach is not entirely zero-shot, since it still requires a small labeled development set for adjusting the $\alpha$ parameter.

## Ethical Considerations

To the best of our knowledge, the paper does not pose any immediate ethical concerns.

## Acknowledgments

We thank the students in the Center for Language and Speech Technologies (CLSP) for their insightful feedback. The authors would like to thank anonymous reviewers for their constructive feedback. This project is supported by generous gifts from Johns Hopkins University, Allen Institute for AI, and Amazon. GPU machines for conducting experiments were provided by ARCH Rockfish cluster (https://www.arch.jhu.edu).

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

# Supplementary Material

## A  Sen and MI are approximations (surrogates) of prompt loss

In this section, we demonstrate that Sen (Sorensen et al., 2022) and MI (Chen et al., 2022) of prompt $p$ on dataset $\mathcal{D}$ are essentially surrogates for the prompt loss $\mathcal{L}$ on $\mathcal{D}$.

**Mutual Information**  Sorensen et al. (2022) hypothesizes that a prompt $p_i$ with higher mutual information (MI) will align a language model to a task better. In prompt selection, MI select the prompt $\hat{p} = \operatorname{argmax}_p \{I\left(f_\theta(\mathcal{D} \circ p); \mathcal{Y}\right)\}$ and MI can be estimated as:

$$
\begin{aligned}
MI(\mathcal{D}, p; \theta) &= I\left(f_\theta(\mathcal{D}, p); \mathcal{Y}\right) \\
&= H(\mathcal{Y}) - H\left(f_\theta(\mathcal{D}, p)\right)
\end{aligned}
\tag{9}
$$

where $H$ refers to entropy, and each term is estimated in expectation using N draws $x \sim \mathcal{D}_X$:

$$
H(\mathcal{Y}) = H\left(\frac{1}{N} \sum_{x \in \mathcal{D}_X} f_\theta(x \circ p)\right)
\tag{10}
$$

$$
H\left(\mathcal{Y} \mid f_\theta(\mathcal{D}, p)\right) = \frac{1}{N} \sum_{x \in \mathcal{D}_X} H(f_\theta(x \circ p))
$$

According to the Weak Law of Large Numbers (and assume that test samples $x \in \mathcal{D}$ are independently drawn from an unknown distribution $P_\mathcal{D}$), it is easy to obtain that

$$
\lim_{|\mathcal{D}| \to \infty} \mathbb{P}\left(|H(\mathcal{Y}) - H(\mathbf{E})| \geq \epsilon\right) = 0
\tag{11}
$$

Where $\mathbf{E}$ refers to the expectation $\mathbb{E}_{x \in P_\mathcal{D}} P(Y \mid x \circ p; \theta)$, and it is a fixed distribution once $f$ and $P_\mathcal{D}$ are determined. As shown by Equation 11, $H(\mathcal{Y})$ converges to a constant as the test sample number increases. Now we focus on the second term of $MI(\mathcal{D}, p)$, as shown in Equation 9. We also re-write it as follows:

$$
\begin{aligned}
H\left(\mathcal{Y} \mid f_\theta(\mathcal{D}, p)\right) &= \frac{1}{|\mathcal{D}|} \sum_{x \in \mathcal{D}_X} H(f_\theta(x \circ p)) \\
&= \frac{1}{|\mathcal{D}|} \sum_{x \in \mathcal{D}_X} [\ell_{ce}(y, f_\theta(x \circ p)) \\
&\quad - \mathrm{KL}(f_\theta(x \circ p) \| y)] \\
&= \mathcal{L}(p, \mathcal{D}, \theta) - \frac{1}{|\mathcal{D}|} \sum_{x \in \mathcal{D}_X} \mathrm{KL}(f_\theta(x \circ p) \| y)
\end{aligned}
$$

where $\ell_{ce}$ refers to cross-entropy and KL is the KL divergence. The equation above illustrates a relation between the second term of $MI$ and prompt loss. The gap between MI and prompt loss is the average KL

divergence. Overall, MI can be formulated as follows:

$$\mathrm{MI}(\mathcal{D}, p) = H(\mathbf{E}) - \mathcal{L}(p, \mathcal{D}, \theta) \tag{12}$$

$$+ \frac{1}{|\mathcal{D}|} \sum_{x \in \mathcal{D}_X} \mathrm{KL}(f_\theta(x \circ p) || y) \tag{13}$$

Equation 12 shows that maximizing mutual information is equivalent to minimizing prompt loss to a certain degree, indicating that MI serves as a surrogate for prompt loss.

**Sensitivity**  Sensitivity (Sen) reflects how much the model output changes given small perturbations of the input. Sen first creates a perturbed prompt set $\mathcal{P}$ given a prompt $p$, by changing demo order $\sigma$ and adding perturbation $\epsilon$ to the prompt instruction $I$. We direct readers to the original paper (Chen et al., 2022) for details of how such prompt sets can be created. Sensitivity on one single test sample $x$ is formally denoted as follows:

$$\mathrm{SEN}(x, p) = \sum_{p' \in \mathcal{P}} \mathbf{1} \left[ f_\theta(x \circ p) \neq f_\theta(x \circ p') \right] \tag{14}$$

Naturally, we can extend this sample-level metric to the dataset level. Given test samples $\mathcal{D}$, the SEN of prompt $p$ is defined as follows:

$$\mathrm{SEN}(\mathcal{D}, p) = \frac{1}{N} \sum_{x \in \mathcal{D}_X} \mathrm{SEN}(x, p) \tag{15}$$

We can re-write the formula for Sen as follows:

$$
\begin{aligned}
\mathrm{SEN}(\mathcal{D}, p) &= \frac{1}{N} \sum_{x \in \mathcal{D}_X} \mathrm{SEN}(x, p) \\
&= \frac{1}{|\mathcal{D}|} \sum_{x \in \mathcal{D}_X} \mathbf{E}_{p'} \ell_{01}(f_\theta(x \circ p'), f_\theta(x \circ p)) \\
&= \mathcal{L}(p, \mathcal{D}, \theta) - \frac{1}{|\mathcal{D}|} \sum_{x \in \mathcal{D}_X} \mathbf{E}_{p'} \ell_{01}(f_\theta(x \circ p'), y)
\end{aligned}
$$

Note that $\mathcal{L}(p, \mathcal{D})$ is a 0-1 loss instead of cross-entropy loss as shown in MI's derivation. The equation above shows that SEN can be regarded as a surrogate for the prompt loss $\mathcal{L}$. Therefore, minimizing SEN is partially equal to minimizing prompt loss, explaining why a low-sensitivity prompt achieves better performance, as empirically verified by Chen et al. (2022).

Generally, the gap between prompt loss $\mathcal{L}$ and two surrogates (MI and SEN) is determined by the distance (i.e., KL divergence) between the model's prediction $f_\theta(x \circ p)$ distribution and ground-truth label. When $f_\theta(x)$ is identical to round-truth label, MI and SEN become perfect surrogates for prompt loss $\mathcal{L}$.

## B  On the approximation gap of flatness and $\mathcal{F}$

This section details the approximation gap of flatness PFLAT towards $\mathcal{F}$. Firstly, we recall the definition of PFLAT and $\mathcal{F}$.

$$
\begin{aligned}
\mathrm{PFLAT}(p, \mathcal{D}, \theta) &= \frac{1}{|\mathcal{D}|} \sum_{x \in \mathcal{D}_X} \mathbf{E}_\epsilon [\ell(f_\theta(p \circ x), f_{\theta + \epsilon_1}(p \circ x)) \\
&\quad - \ell(f_\theta(p \circ x), f_{\theta + \epsilon_2}(p \circ x))] \\
&= \frac{1}{|\mathcal{D}|} \sum_{x \in \mathcal{D}_X} \mathbf{E}_\epsilon \| \ell(f_\theta(p \circ x), f_{\theta + \epsilon}(p \circ x)) \|_2
\end{aligned}
\tag{16}
$$

Also, we re-write $\mathcal{F}(p, \mathcal{D}, \theta)$ as follows:

$$\mathcal{F}(p, \mathcal{D}, \theta) = \| \nabla_\theta \mathcal{L}(p, \mathcal{D}, \theta) \|_2 \tag{17}$$

$$= \frac{1}{|\mathcal{D}|} \sum_{x,y \in \mathcal{D}} \nabla_\theta \| \ell(f_\theta(p \circ x), y) \|_2 \tag{18}$$

Thus, the approximation gap can be obtained through [Equation 16](#) and [Equation 17](#). When the model's confidence is identical to ground-truth labels, PFLAT is a precise approximator of $\mathcal{F}$.

## C Results on correlation

Here are the full results of correlation comparisons in our paper, as shown in Table 2.

| Model | Methods | AGNews | | CB | | DBpedia | | SST-2 | | RTE | |
|---|---|---|---|---|---|---|---|---|---|---|---|
| | | Pr | Spr | Pr | Spr | Pr | Spr | Pr | Spr | Pr | Spr |
| GPT2-base | MI | 21.9 | 22.5 | 3.5 | 4.5 | 30.1 | 25.1 | 19.2 | 16.8 | 20.6 | 23.5 |
| | Sen | 8.6 | 7.6 | 14.3 | 13.4 | -10.6 | -14.2 | 5.6 | 3.4 | -9.9 | -13.9 |
| | pFLAT | 21.4 | 19.8 | -9.1 | -8.1 | 21.3 | 22.0 | 18.3 | 20.2 | 20.2 | 20.4 |
| | MI+Sen | 22.0 | 16.4 | 17.1 | 18.1 | 26.0 | 23.7 | 20.1 | 21.2 | 24.9 | 23.8 |
| | MI+pFLAT | 26.3 | 25.7 | 20.5 | 20.7 | 24.1 | 25.6 | 23.4 | 21.7 | 15.4 | 17.5 |
| | Sen+pFLAT | 28.6 | 29.3 | 28.1 | 26.7 | 29.4 | 28.1 | 23.6 | 24.5 | 27.2 | 25.3 |
| GPT2-medium | MI | 27.5 | 26.4 | 26.7 | 23.0 | 28.9 | 26.9 | 27.1 | 25.0 | 22.5 | 16.2 |
| | Sen | -11.2 | 3.5 | -4.5 | -7.7 | -8.6 | -10.8 | 10.1 | 5.4 | -8.4 | -10.3 |
| | pFLAT | 23.8 | 26.0 | 21.6 | 22.5 | 20.6 | 23.4 | 23.8 | 23.3 | 20.2 | 17.7 |
| | MI+Sen | 24.7 | 22.8 | 18.7 | 20.4 | 23.7 | 27.0 | 27.7 | 26.5 | 11.2 | 13.0 |
| | MI+pFLAT | 29.0 | 30.1 | 22.9 | 20.5 | 29.9 | 31.9 | 27.0 | 28.7 | 25.7 | 20.6 |
| | Sen+pFLAT | 28.1 | 29.0 | 23.6 | 26.6 | 33.1 | 31.8 | 32.3 | 32.7 | 24.0 | 17.3 |
| GPT2-large | MI | 23.4 | 21.0 | 20.9 | 20.9 | 24.2 | 27.2 | 20.2 | 21.3 | 22.8 | 18.6 |
| | Sen | 11.0 | 5.6 | -6.0 | -4.3 | -5.9 | -8.9 | -6.7 | 5.8 | 11.0 | 22.1 |
| | pFLAT | 20.0 | 20.7 | 19.4 | 22.4 | 21.7 | 22.5 | 20.1 | 18.3 | 22.1 | 20.2 |
| | MI+Sen | 25.0 | 21.3 | 24.1 | 24.6 | 23.0 | 27.0 | 25.3 | 24.0 | 19.4 | 21.5 |
| | MI+pFLAT | 25.4 | 26.7 | 25.4 | 29.3 | 26.3 | 27.6 | 30.1 | 28.9 | 35.3 | 30.9 |
| | Sen+pFLAT | 29.5 | 28.8 | 28.0 | 28.5 | 24.9 | 28.4 | 31.3 | 30.4 | 19.3 | 20.9 |
| GPT2-xl | MI | 22.7 | 24.3 | 18.9 | 20.1 | 24.7 | 22.0 | 15.3 | 16.8 | 14.7 | 21.2 |
| | Sen | 5.6 | 9.9 | -3.8 | -5.0 | -8.4 | -13.8 | 10.1 | 2.4 | -4.3 | -10.1 |
| | pFLAT | 10.6 | 6.2 | 20.1 | 18.1 | 23.7 | 23.4 | 14.2 | 12.0 | 24.2 | 26.9 |
| | MI+Sen | 20.5 | 19.2 | 18.2 | 21.0 | 21.9 | 22.0 | 20.1 | 19.7 | 18.0 | 20.4 |
| | MI+pFLAT | 21.4 | 20.5 | 22.3 | 19.8 | 25.1 | 26.9 | 22.3 | 20.9 | 16.4 | 18.1 |
| | Sen+pFLAT | 25.3 | 21.8 | 24.3 | 25.7 | 25.3 | 22.3 | 24.1 | 20.0 | 25.3 | 25.5 |

Table 2: Pearson (Pr) and Spearman (Spr) correlation between prompts' performance and the metrics of various method. Overall, flatness-based metrics obtain higher correlations. Red means the best performance.

# D   Results on Prompt Retrieval

Here are the full results of prompt retrieval performance in our paper, as shown in Table 3 and Table 4.

| Model | Methods | SST-2 | | | RTE | | | TREC | | |
|---|---|---|---|---|---|---|---|---|---|---|
| | | N@1 | N@3 | Rate | N@1 | N@3 | Rate | N@1 | N@3 | Rate |
| GPT2-base | MI | 54.1 | 55.0 | 83.7 | 56.1 | 51.5 | 87.8 | 53.6 | 54.8 | 83.9 |
| | MI+PFLAT | 56.6 | 59.4 | 90.8 | 48.2 | 49.4 | 92.1 | 55.8 | 56.7 | 91.2 |
| | Sen | 52.2 | 53.6 | 83.1 | 55.5 | 51.3 | 86.5 | 53.8 | 55.0 | 83.4 |
| | Sen+PFLAT | 49.2 | 59.9 | 92.7 | 69.1 | 57.8 | 92.4 | 54.7 | 56.0 | 93.3 |
| GPT2-medium | MI | 43.2 | 48.7 | 81.9 | 43.5 | 48.7 | 85.9 | 52.9 | 57.6 | 86.0 |
| | MI+PFLAT | 51.6 | 52.2 | 91.8 | 56.6 | 56.2 | 92.8 | 58.0 | 60.1 | 94.0 |
| | Sen | 45.1 | 49.0 | 82.7 | 44.6 | 55.2 | 88.7 | 54.0 | 56.0 | 86.4 |
| | Sen+PFLAT | 54.1 | 57.9 | 90.3 | 57.1 | 57.9 | 93.3 | 57.4 | 57.9 | 95.2 |
| GPT2-large | MI | 47.5 | 44.9 | 81.0 | 36.6 | 40.6 | 90.2 | 56.1 | 54.6 | 88.7 |
| | MI+PFLAT | 51.2 | 56.8 | 90.4 | 64.1 | 42.3 | 96.0 | 58.1 | 54.2 | 93.4 |
| | Sen | 50.2 | 51.7 | 77.5 | 51.1 | 47.5 | 86.5 | 49.9 | 53.2 | 87.9 |
| | Sen+PFLAT | 56.4 | 59.0 | 91.7 | 60.9 | 56.3 | 97.6 | 57.0 | 56.8 | 94.9 |
| GPT2-xl | MI | 52.1 | 51.2 | 86.1 | 22.7 | 28.4 | 87.7 | 54.6 | 55.0 | 85.6 |
| | MI+PFLAT | 52.0 | 53.5 | 95.5 | 44.6 | 42.6 | 97.9 | 51.4 | 53.0 | 95.3 |
| | Sen | 47.7 | 53.1 | 87.5 | 23.6 | 26.8 | 86.4 | 52.8 | 53.4 | 85.0 |
| | Sen+PFLAT | 57.3 | 54.2 | 95.0 | 32.9 | 36.5 | 96.2 | 56.6 | 53.4 | 95.0 |

Table 3: Results of high-performance prompts retrieval, we can see that metric combined prompt loss and flatness achieve better performance. Specifically, N@1 represents NDCG@1. Red means the best performance.

| Model | Methods | AGNews | | | CB | | | DBpedia | | |
|---|---|---|---|---|---|---|---|---|---|---|
| | | N@1 | N@3 | Rate | N@1 | N@3 | Rate | N@1 | N@3 | Rate |
| GPT2-base | MI | 49.2 | 50.4 | 86.8 | 31.5 | 42.6 | 88.2 | 39.9 | 47.8 | 86.6 |
| | Sen | 46.5 | 56.8 | 85.4 | 34.3 | 50.0 | 85.6 | 35.9 | 46.3 | 86.7 |
| | MI+PFLAT | 52.1 | 48.9 | 91.9 | 43.1 | 44.3 | 94.6 | 39.9 | 52.8 | 94.2 |
| | Sen+PFLAT | 52.3 | 54.0 | 92.8 | 34.8 | 45.4 | 95.1 | 50.1 | 48.4 | 94.8 |
| GPT2-medium | MI | 46.8 | 50.4 | 88.2 | 57.8 | 48.5 | 86.2 | 51.3 | 50.0 | 86.6 |
| | Sen | 44.3 | 56.8 | 85.9 | 64.4 | 60.6 | 86.0 | 53.3 | 52.0 | 87.1 |
| | MI+PFLAT | 51.9 | 58.9 | 93.9 | 53.7 | 47.0 | 95.7 | 51.3 | 58.4 | 95.2 |
| | Sen+PFLAT | 50.8 | 54.0 | 94.8 | 63.7 | 52.0 | 94.5 | 56.0 | 55.0 | 95.6 |
| GPT2-large | MI | 53.2 | 51.2 | 87.3 | 34.1 | 44.8 | 85.9 | 53.3 | 55.3 | 87.0 |
| | Sen | 46.9 | 50.6 | 84.0 | 28.8 | 40.1 | 83.1 | 33.8 | 43.1 | 84.8 |
| | MI+PFLAT | 47.0 | 45.9 | 95.2 | 37.1 | 50.1 | 96.3 | 50.9 | 49.1 | 96.3 |
| | Sen+PFLAT | 52.1 | 53.8 | 94.7 | 24.1 | 46.2 | 95.9 | 39.6 | 54.8 | 97.1 |
| GPT2-xl | MI | 44.5 | 60.8 | 88.6 | 51.4 | 62.3 | 86.1 | 55.7 | 53.0 | 85.7 |
| | Sen | 48.1 | 57.0 | 87.8 | 48.8 | 53.0 | 83.1 | 44.1 | 52.9 | 84.1 |
| | MI+PFLAT | 48.9 | 46.3 | 97.4 | 53.6 | 69.2 | 96.4 | 58.7 | 49.7 | 96.0 |
| | Sen+PFLAT | 53.0 | 57.1 | 96.0 | 54.8 | 54.1 | 96.0 | 47.7 | 58.4 | 96.2 |

Table 4: Results of high-performance prompts retrieval on AGNews, CB, and DBpedia. we can see that metric combined prompt loss and flatness achieve better performance. Red means the best performance.

# E    Comparison to Automatic Prompt Generation Algorithms

Here we compare PFLAT to automatic prompt-generation, namely RLPrompt (Deng et al., 2022), Tempera (Zhang et al., 2023a), and GrIPs (Prasad et al., 2023). The primary objective of these algorithms is to automatically generate prompts that would be apt for prompt tuning. By contrast, our study aims to scrutinize and identify a prompt that would be advantageous for ICL. In this section, we present empirical results based on prompts produced by off-the-shelf models of RLPrompt, Tempera, GrIPs. Then we compare the performance of the prompts obtained via various approaches, including RLPrompt, Tempera, GrIPs, and our method (Sen+PFLAT), as depicted in Table 5. The results illustrate that the prompt selected by our method exhibits superior ICL performance. Besides, we show some examples of RLPrompt, Tempera, and GrIPs in Table 6, Table 7, Table 8.

| Model | Methods | SST-2 | | | RTE | | | TREC | | |
|-------|---------|--------|--------|--------|--------|--------|--------|--------|--------|--------|
| | | 1-shot | 4-shot | 8-shot | 1-shot | 4-shot | 8-shot | 1-shot | 4-shot | 8-shot |
| GPT2-xl | RLPrompt | 54.1 | 56.0 | 60.7 | 52.1 | 54.5 | 57.8 | 25.6 | 27.8 | 30.9 |
| | Tempera | 55.0 | 59.4 | 61.8 | 52.2 | 58.4 | 59.1 | 24.8 | 28.7 | 31.2 |
| | GrIPs | 52.2 | 58.6 | 60.1 | 51.5 | 53.3 | 56.5 | 23.8 | 25.0 | 27.9 |
| | Sen+PFLAT | 58.9 | 63.9 | 65.7 | 55.6 | 58.9 | 61.9 | 29.7 | 32.1 | 34.7 |

Table 5: In-context learning performance of prompts from different methods. We can observe that the prompt selected by our method achieves better in-context learning performance.

---

**Examples of prompt generated by RLPrompt, Tempera, and GrIPs**

- **[RLPrompt]**: Sentiment of the sentence is negative or positive.

- **[Tempera]**: Given text, given text, Classify whether it is good or bad.

- **[GrIPs]**: Your task as "positive" or "negative".

---

Table 6: Instructions from RLPrompt, Tempera, and GrIPs for SST-2 task

---

**Examples of prompt generated by RLPrompt, Tempera, and GrIPs**

- **[RLPrompt]**: premise follow that hypo yes or no?

- **[Tempera]**: Given premise, does it follow hypothesis?

- **[GrIPs]**: Does the information support premise?

---

Table 7: Instructions from RLPrompt, Tempera, and GrIPs for RTE task

---

**Examples of prompt generated by RLPrompt, Tempera, and GrIPs**

- **[RLPrompt]**: The topic of the question is

- **[Tempera]**: Given the info, what's the topic

- **[GrIPs]**: Topic of the sentence

---

Table 8: Instructions from RLPrompt, Tempera, and GrIPs for TREC task

## F  Instructions

Here we include the pool of natural language prompts (instructions) used in each task. We list instructions for SST-2 in Table 9, RTE in Table 10, TREC in Table 11, AGNews in Table 12, CB in Table 13 and DBPedia in Table 14.

---

**SST-2 Instructions**

- Suppose we have the following premise, Can we infer that hypothesis? Yes, no, or maybe?

- Based on the previous premise, is it true for the hypothesis?

- See on the following information, is the claim right?

- Given that premise, does it follow that hypothesis? Yes, no, or maybe?

- Given the premise, are we justified in saying that hypothesis? Yes, no, or maybe?

- Based on the text, question: hypothesis is True, False, or Neither?

- Keeping in mind the above text, consider: hypothesis is always, sometimes, or never correct?

- Given premise. Is it guaranteed true that hypothesis? Yes, no, or maybe?

- Given that premise. Therefore, it must be true that hypothesis? Yes, no, or maybe?

- Assume it is true that premise. Therefore, hypothesis is guaranteed, possible, or impossible?

- Using only the following description and what you know about the world, hypothesis is definitely correct, incorrect, or inconclusive?

- Take the following as truth. Then the hypothesis is true, false, or inconclusive?

- Can we derive that hypothesis if we have the following premise? Yes, no, or perhaps?

- Can we arrive at that conclusion if we possess the following information? Possibly, no, or both?

- Does that premise flow from the given premise? Yes, no, or perhaps?

- Does that information support the claim?

- Is the assertion accurate in light of such information?

- Considering the text, which of the following statements is True, False, or Both?

- Think about the question: Is hypothesis always, occasionally, or never correct?

- Can we derive that conclusion if we have the following information? Yes, no, or possibly?

---

Table 9: Instructions for SST-2 task

**RTE Instructions**

- Using only the above description and what you know about the world, is hypothesis definitely correct? Yes or no?

- Given premise, Is it guaranteed true that hypothesis? Yes or no?

- Suppose premise, Can we infer that hypothesis? Yes or no?

- Given premise Should we assume that hypothesis is true? Yes or no?

- Given that premise, Does it follow that hypothesis Yes or no?

- Given premise. Is it guaranteed true that hypothesis? Yes, no, or maybe?

- Given that premise. Therefore, it must be true that hypothesis? Yes, no, or maybe?

- Assume it is true that premise. Therefore, hypothesis is guaranteed, possible, or impossible?

- Using only the following description and what you know about the world, hypothesis is definitely correct, incorrect, or inconclusive?

- Take the following as truth. Then the hypothesis is true, false, or inconclusive?

- Can we derive that hypothesis if we have the following premise? Yes, no, or perhaps?

- Can we arrive at that conclusion if we possess the following information? Possibly, no, or both?

- Does that premise flow from the given premise? Yes, no, or perhaps?

- Does that information support the claim?

- Is the assertion accurate in light of such information?

- Considering the text, which of the following statements is True, False, or Both?

- Think about the question: Is hypothesis always, occasionally, or never correct?

- Can we derive that conclusion if we have the following information? Yes, no, or possibly?

- Suppose we have the following premise, Can we infer that hypothesis? Yes, no, or maybe?

- Based on the previous premise, is it true for the hypothesis?

Table 10: Instructions for RTE task

**TREC Instructions**

- What kind of label best describes this question below?

- What is this a piece of question regarding for?

- What is the category of the following question?

- Which is the most relevant topic of the following question?

- Give the topic of the given question.

- Read the question below, provide its focused topic.

- Is this a piece of question regarding ABBR, ENTY, DESC, HUM, LOC, or NUM?

- Which section of a newspaper would this question likely appear in?

- What label would you use to characterize this question item?

- What term can best sums up this question?

- Which category most accurately sums up this question item?

- What label would you use to characterize this question?

- Is this question related to ABBR, ENTY, DESC, HUM, LOC, or NUM?

- Does this question story have anything to do with ABBR, ENTY, DESC, HUM, LOC, or NUM?

- Read the question below and explain its specific subject.

- Please read the following material and explain its main point.

- Provide your thoughts on the content below after reading it.

- Describe the question's subject as follows.

- For what purpose does this question item exist?

- Are there any ABBR, ENTY, DESC, HUM, LOC, or NUM related stories in this question?

Table 11: Instructions for TREC task

**AGNews Instructions**

- What label best describes this news article?

- What is this a piece of news regarding for?

- What is the category of the following news?

- Which is the most relevant topic of the following news?

- Give the topic of the given text.

- Read the text below, provide its focused topic.

- Is this a piece of news regarding world, sport, business,or science?

- Which section of a newspaper would this article likely appear in?

- What label would you use to characterize this news item?

- What term best sums up this news report?

- Which category most accurately sums up this news item?

- What label would you use to characterize this news story?

- Is this news related to the world, sports, business, or science?

- Does this news story have anything to do with the world, sports, business, or science?

- Read the paragraph below and explain its specific subject.

- Please read the following material and explain its main point.

- Provide your thoughts on the content below after reading it.

- Describe the text's subject as follows.

- For what purpose does this news item exist?

- Are there any world-related, sports, business, or science-related stories in this news?

Table 12: Instructions for AGNews task

**CB Instructions**

- Suppose we have the following premise, Can we infer that hypothesis? Yes, no, or maybe?

- Based on the previous premise, is it true for the hypothesis?

- See on the following information, is the claim right?

- Given that premise, does it follow that hypothesis? Yes, no, or maybe?

- Given the premise, are we justified in saying that hypothesis? Yes, no, or maybe?

- Based on the text, question: hypothesis is True, False, or Neither?

- Keeping in mind the above text, consider: hypothesis is always, sometimes, or never correct?

- Given premise. Is it guaranteed true that hypothesis? Yes, no, or maybe?

- Given that premise. Therefore, it must be true that hypothesis? Yes, no, or maybe?

- Assume it is true that premise. Therefore, hypothesis is guaranteed, possible, or impossible?

- Using only the following description and what you know about the world, hypothesis is definitely correct, incorrect, or inconclusive?

- Take the following as truth. Then the hypothesis is true, false, or inconclusive?

- Can we derive that hypothesis if we have the following premise? Yes, no, or perhaps?

- Can we arrive at that conclusion if we possess the following information? Possibly, no, or both?

- Does that premise flow from the given premise? Yes, no, or perhaps?

- Does that information support the claim?

- Is the assertion accurate in light of such information?

- Considering the text, which of the following statements is True, False, or Both?

- Think about the question: Is hypothesis always, occasionally, or never correct?

- Can we derive that conclusion if we have the following information? Yes, no, or possibly?

Table 13: Instructions for CB task

**DBPedia Instructions**

- What label best describes this paragraph?

- What is this paragraph regarding for?

- What is the category of the following paragraph?

- Which is the most relevant topic of the following paragraph?

- Give the topic of the given text.

- Read the text below, provide its focused topic.

- Is this paragraph regarding company, educational institution, artist, athlete, office holder, mean of transportation, building, natural place, village, animal, plant, album, film or written work?

- What label would you use to characterize this paragraph?

- What term best sums up this paragraph?

- Which category most accurately sums up this paragraph?

- What label would you use to characterize this paragraph?

- Is this paragraph related to company, educational institution, artist, athlete, office holder, mean of transportation, building, natural place, village, animal, plant, album, film or written work?

- Does this news story have anything to do with company, educational institution, artist, athlete, office holder, mean of transportation, building, natural place, village, animal, plant, album, film or written work?

- Read the paragraph below and explain its specific subject.

- Please read the following material and explain its main point.

- Describe the text's subject as follows.

- Are there any company, educational institution, artist, athlete, office holder, mean of transportation, building, natural place, village, animal, plant, album, film or written work content in this paragraph?

- Given a list of categories: company, educational institution, artist, athlete, office holder, mean of transportation, building, natural place, village, animal, plant, album, film or written work, what category does the paragraph belong to?

- Pick one category for the following text. The options are - company, educational institution, artist, athlete, office holder, mean of transportation, building, natural place, village, animal, plant, album, film or written work.

- Given a choice of categories company, educational institution, artist, athlete, office holder, mean of transportation, building, natural place, village, animal, plant, album, film or written work, the text refers to which one?

Table 14: Instructions for DBPedia task