# OpenReview forum: "Flatness-Aware Prompt Selection Improves Accuracy and Sample Efficiency"
_EMNLP/2023/Conference — EMNLP 2023 Findings_

### Official Review · Reviewer_TSnL · 2023-08-03

**Soundness:** 3

**Excitement:**

2: Mediocre: This paper makes marginal contributions (vs non-contemporaneous work), so I would rather not see it in the conference.

**Paper Topic And Main Contributions:**

Prompt selection is essential for prompt tuning for model alignment of LLMs. This paper introduces PFLAT (prompt flatness), a new metric to quantify the “expected utility” of a language prompt. This metric is inspired by flatness regularization in statistical learning that quantifies the robustness of the model towards its parameter perturbations. Empirically,
combining PFLAT with existing metrics improves both performance and sample efficiency.


**Questions For The Authors:**

Questions:
1.	How to deep understand the prompts listed in table 9 to 14? Can you give some comparison of “list of prompts selected before using pFlat” vs. “after using pFloat” and the related examples of classification tasks? Currently it is still difficult to have a clear learning of the detailed connections between pFlat and the tables 9 to 14.
2.	It seems that your approach can be applied to generation tasks in a smooth way and what blocked your continue experiments? Considering they are not that difficult and there are rich generation-based datasets for model alignment.


**Reasons To Accept:**

Strong:
1.	A new framework for prompt selection that merges prompt loss and flatness, enabling the integration of previous studies to elucidate their distinctions and efficacy.
2.	Better performances under a list of classification NLP tasks.


**Reasons To Reject:**

Weak:
1.	Authors showed experiments for classification tasks only and prefer to learn the results for more challenging tasks, generation tasks, in this paper, not future work.
2.	Not quite clear about the relationship between the listed prompts in table 9 to table 14 with the classification task examples. Currently it is difficult to learn why these prompts are more important evaluated by pFlat.


**Reproducibility:**

3: Could reproduce the results with some difficulty. The settings of parameters are underspecified or subjectively determined; the training/evaluation data are not widely available.

**Reviewer Confidence:**

3: Pretty sure, but there's a chance I missed something. Although I have a good feel for this area in general, I did not carefully check the paper's details, e.g., the math, experimental design, or novelty.

---

> ### Author Rebuttal · Authors · 2023-08-29
>
> Dear reviewer,
>
> We deeply appreciate the time and effort you have dedicated to reviewing this paper.
>
> **Q1**: How to deep understand the prompts listed in table 9 to 14? Can you give some comparison of “list of prompts selected before using pFlat” vs. “after using pFloat” and the related examples of classification tasks? Currently it is still difficult to have a clear learning of the detailed connections between pFlat and the tables 9 to 14.
>
> **A1**: The prompts listed in Tables 9-14 are candidate prompts crafted by humans, and this is some prior/agnostic to any evaluation on them.
>
> As for comparisons between “prompts selected before using pFlat” vs. “after using pFlat”: empirically we observed that better prompts (selected by pFlat + x) generally (but not always) contain more specific details. As an example for the AGNews,
>
>  - Prompt selected with MI metric: "Which section of a newspaper would this article most likely be found in?"
>  - Prompt selected with MI+Flat metric: "Is this news article related to topics like the world, sports, business, or science?”
>
> We acknowledge that it is not always clear why certain prompts lead to higher prompt-selection scores and hence, there is need for more research on interpretability of such scores in future work.
>
> We appreciate the feedback and will incorporate these qualitative findings into the next version of our paper.
>
> **Q2**: It seems that your approach can be applied to generation tasks in a smooth way and what blocked your continued experiments? Considering they are not that difficult and there are rich generation-based datasets for model alignment.
>
> **A2**: There are several reasons why focused on classification tasks:
>  - Comparison with prior work: Previous works (Sorensen et al., 2022, Chen et al., 2022) like sensitivity and mutual information conduct experiments across classification tasks and adapting them to non-classification tasks is non-trivial. To have objective comparison with these works, we were compelled to adopt classification tasks.
>  - Difficulty of evaluation: automatic evaluation of generation tasks remains challenging and prone to various errors. Even human evaluation (e.g., through MTurk) can be quite noisy and non-trivial for many open-ended generation tasks such as summarization.
>
> With all these said, we agree that the overall field should move on to more open-ended generation tasks. If you feel that there is an obvious benchmark (that may accept reliable automatic or human-driven evaluation) that we should evaluate for our revision, please flag it for us and we are happy to include it.

---

### Official Review · Reviewer_CMk4 · 2023-08-04

**Soundness:** 4

**Excitement:**

4: Strong: This paper deepens the understanding of some phenomenon or lowers the barriers to an existing research direction.

**Paper Topic And Main Contributions:**

This paper explores what makes the existing methods for prompt selection effective and their relationship.  To adress those, this work proposes a formal optimization framework that unifies several existing prompt selection metrics such as MI and SEN. And introduce PFLAT, a metric for selecting prompts that is more robust to LLMs’ parametric perturbations.

**Reasons To Accept:**

This paper provides theoretical foundations for the introduced PFLAT metric and its relationship with other prompt selection metrics, and provides a comprehensive understanding of existing methods. Experiments results demonstrate the effectiveness of the method for prompt selection.

**Reasons To Reject:**

The  proposed method is tested only on classification tasks.

**Reproducibility:**

4: Could mostly reproduce the results, but there may be some variation because of sample variance or minor variations in their interpretation of the protocol or method.

**Reviewer Confidence:**

3: Pretty sure, but there's a chance I missed something. Although I have a good feel for this area in general, I did not carefully check the paper's details, e.g., the math, experimental design, or novelty.

---

> ### Author Rebuttal · Authors · 2023-08-29
>
> Dear reviewer,
>
> We deeply appreciate the time and effort you have dedicated to reviewing this paper.
>
> **Q1**: The proposed method is tested only on classification tasks.
>
> **A1**: There are several reasons why focused on classification tasks:
>  - Comparison with prior work: Previous works (Sorensen et al., 2022, Chen et al., 2022) like sensitivity and mutual information conduct experiments across classification tasks and adapting them to non-classification tasks is non-trivial. To have objective comparison with these works, we were compelled to adopt classification tasks.
>  - Difficulty of evaluation: automatic evaluation of generation tasks remains challenging and prone to various errors. Even human evaluation (e.g., through MTurk) can be quite noisy and non-trivial for many open-ended generation tasks such as summarization.
>
> With all these said, we agree that the overall field should move on to more open-ended generation tasks. If you feel that there is an obvious benchmark (that may accept reliable automatic or human-driven evaluation) that we should evaluate for our revision, please flag it for us and we are happy to include it.

---

### Official Review · Reviewer_wwPW · 2023-08-05

**Soundness:** 4

**Excitement:**

4: Strong: This paper deepens the understanding of some phenomenon or lowers the barriers to an existing research direction.

**Paper Topic And Main Contributions:**

The authors introduce a new metric, PFLAT, and demonstrate that its integration with existing metrics (Mutual Information and Sensitivity) enhances the prompt selection process. Evaluation metrics include correlation with accuracy and ranking rate. Additional analyses illustrate the benefits of PFLAT in continuous prompt selection, significant improvements when the model size is large, and robust performance with low sampling data size.

**Questions For The Authors:**

Evaluation Metrics: Please clarify the rationale behind using correlation with accuracy as an evaluation metric instead of directly utilizing accuracy.

**Reasons To Accept:**

The manuscript is well-composed. The research design is clear and thorough, and the results thoroughly substantiate the claims. The research question is engaging.

**Reasons To Reject:**

None

**Reproducibility:**

5: Could easily reproduce the results.

**Reviewer Confidence:**

3: Pretty sure, but there's a chance I missed something. Although I have a good feel for this area in general, I did not carefully check the paper's details, e.g., the math, experimental design, or novelty.

**Typos Grammar Style And Presentation Improvements:**

* Please insert a new paragraph at line 297 before 'Correlation with accuracy.'
* There is a missing period at line 144.

---

> ### Author Rebuttal · Authors · 2023-08-29
>
> Dear reviewer,
>
> We deeply appreciate the time and effort you have dedicated to reviewing this paper.
>
> **Q1**: Evaluation Metrics: Please clarify the rationale behind using correlation with accuracy as an evaluation metric instead of directly utilizing accuracy.
>
> **A1**: In our experiments, we also employ a metric that quantifies relative accuracy and complements our correlation-based metric (as indicated in the "Rate" metric). However, we use correlation as a complementary metric for direct accuracy due to the following reason:
>
> Intuitively, "correlation" measures the alignment between the scores of prompt-selection metrics, including the one we propose, and the downstream accuracies of each prompt. Essentially, this evaluation contrasts the relative quality of prompts based on their accuracy with their prompt-selection accuracy. A high correlation indicates that this prompt-selection metric can serve as a “surrogate” (proxy) for selecting the most accurate prompt, bypassing the direct maximization of accuracy which often demands extra held-out labeled data.
>
> Overall, while we could have showcased the accuracy of the prompt with the top-most prompt-selection accuracy, such an assessment only gauges the quality of the best (argmax) prompt and overlooks their relative (ranked) quality. We will update the document to elucidate these intuitions further.

---

### Meta-Review · Area_Chair_X5xK · 2023-09-25

**Recommendation:** 3

**Metareview:**

The paper introduces a new metric called PFLAT that enhances prompt selection by integrating it with existing metrics.
Pros:
 It is more robust to LLMs’ parametric perturbations and provides significant improvements for large model sizes.
The paper also provides a comprehensive understanding of existing methods and their relationship with PFLAT.
The experiments demonstrate the effectiveness of the method for prompt selection

Cons:
The paper showed experiments for classification tasks only, no results for more challenging tasks such as generation tasks.
It is not clear what leads to higher prompt-selection scores,  lack of interpretability of the scores.

---

### Decision · Program_Chairs · 2023-10-07

**Decision:**

Accept-Findings

**Comment:**

The paper introduces a new metric called PFLAT that enhances prompt selection by integrating it with existing metrics.
Pros:
 It is more robust to LLMs’ parametric perturbations and provides significant improvements for large model sizes.
The paper also provides a comprehensive understanding of existing methods and their relationship with PFLAT.
The experiments demonstrate the effectiveness of the method for prompt selection

Cons:
The paper showed experiments for classification tasks only, no results for more challenging tasks such as generation tasks.
It is not clear what leads to higher prompt-selection scores,  lack of interpretability of the scores.